# The Synergistic Effect of Acidic Properties and Channel Systems of Zeolites on the Synthesis of Polyoxymethylene Dimethyl Ethers from Dimethoxymethane and Trioxymethylene

**DOI:** 10.3390/nano9091192

**Published:** 2019-08-23

**Authors:** Jianbing Wu, Sen Wang, Haitao Li, Yin Zhang, Ruiping Shi, Yongxiang Zhao

**Affiliations:** 1Engineering Research Center of Ministry of Education for Fine Chemicals, Shanxi University, Taiyuan 030006, China; 2State Key Laboratory of Coal Conversion, Institute of Coal Chemistry, Chinese Academy of Sciences, Taiyuan 030001, China

**Keywords:** zeolites, polyoxymethylene dimethyl ethers, dimethoxymethane, trioxymethylene, Brønsted acid sites, the maximum included sphere, steric constraint

## Abstract

A series of zeolites with different topology structures, including SAPO-34, SUZ-4, ZSM-5, USY, MOR, and beta, were used to synthesize polyoxymethylene dimethyl ethers (PODE*_n_*) from dimethoxymethane (DMM) and trioxymethylene (TOM). The influence of acidic properties and channel systems were studied by activity evaluation, characterization, and theoretical calculation. The results confirmed that pore mouth diameter larger than a TOM molecule was an essential prerequisite for the synthesis of PODE*_n_* over zeolites, and the synergistic effect between medium-strong Brønsted acid sites (Brønsted MAS) and the maximal space of zeolites available determined the catalytic performance of all studied zeolites. DMM and TOM were firstly decomposed into methoxymethoxy groups (MMZ) and monomer CH_2_O over Brønsted MAS. Subsequently, the steric constraint of the maximum included sphere, with an appropriate size in zeolite channels, can promote the combination of CH_2_O and MMZ to form transition species ZO(CH_2_O)_n_CH_3_, which reacted with the methyl-end group to form PODE*_n_* over Brønsted MAS. Moreover, the reaction temperature showed different effects on the product selectivity and distribution, which also mainly depends on the size of the maximum space available in zeolite channels.

## 1. Introduction

The growing consumption of petroleum fuels is one of the major reasons for air pollution and environmental imbalance. Many efforts have been made to reduce exhaust gas and solid particulate matter from commercial vehicles. The identification of a highly efficient and clean fuel additive that could enhance diesel oil combustion efficiency and reduce pollutant emissions would be an economical and practical solution. Recently, polyoxymethylene dimethyl ethers (PODE*_n_*, CH_3_–O–(CH_2_–O)*_n_*–CH_3_, *n* > 1) have exhibited much potential as diesel additives, due to their high oxygen content, cetane number, good miscibility, and absence of C–to–C bonds [1,2,3,4,5]. Compared with CH_3_OH (MeOH), CH_3_OCH_3_ (DME), CH_3_OCH_2_OCH_3_ (DMM), and CH_3_OCOOCH_3_ (DMC), PODE*_n_*, especially PODE_2–8_, can directly blend into traditional diesel without any fuel system modifications and can reduce the release of NO_x_ and particulate emissions effectively [6,7,8]. Accordingly, the synthesis of PODE*_n_* has gained much research interest.

PODE*_n_* molecules are primarily made up of a methyl-end group and oxymethylene group, which are usually obtained from methanol (MeOH) or dimethoxymethane (DMM) and formaldehyde (FA) or polyformaldehydes (PF*_n_*), respectively [4]. A previous investigation of the synthesis of PODE*_n_* was carried out in homogeneous catalytic systems with H_2_SO_4_ or CF_3_SO_3_H as catalysts, which resulted in high corrosivity, separation difficulties, and potential environmental damage [9]. Consequently, various heterogeneous catalysts, including ion exchange resins [10,11], solid oxides [12,13,14], and zeolites [15,16,17,18,19,20], were evaluated for PODE*_n_* synthesis, due to their advantages in tunable acidity and separation properties.

Our previous work found that the MCM-22 zeolite can catalyze MeOH and trioxymethylene (TOM) to synthesize PODE*_n_*, with a PODE_3–8_ selectivity of 29% at 120 °C [17]. However, due to the water and additional byproducts that were produced with MeOH and TOM or PF as raw materials, which are detrimental to the desired product properties [21], PODE*_n_* synthesis with DMM and TOM was also studied in detail. A PODE_2–8_ selectivity of 88.5% and a TOM conversion of 85.3% were obtained at 120 °C over the ZSM-5 catalyst with a Si/Al molar ratio of 580 [15]. In general, the synthesis of PODE*_n_* from DMM and TOM is most likely the result of the following reversible reactions (Figure 1): The TOM ring decomposes to monomer CH_2_O, after which CH_2_O is inserted into DMM one by one to generate PODE*_n_* [4]. Although different zeolites have been developed to synthesize PODE*_n_* [15,17,22], most research has mainly focused on acidity regulation, whereas the influence of zeolite frameworks and synergistic effect between acidic properties and channel systems remains to be understood.

The aim of the present investigation was to evaluate the effects of zeolite acidic characteristics and channel systems on the synthesis of PODE*_n_* from DMM and TOM, as well as to further evaluate the synergistic effect between these two factors. SAPO-34, SUZ-4, ZSM-5, MOR, USY, and beta zeolites were selected as catalysts, on the basis of similar Si/Al ratios and different framework types. Experiments were carried out to determine the effects of acidic characteristics and channel systems at different temperatures on the feedstock conversion, product selectivity, and product distribution. The observed effects are interpreted in the light of a series of activity evaluations, characterizations, and theoretical calculations.

## 2. Materials and Methods

Boehmite(P-D-03) was purchased from Aluminum Corporation of China. Carbon-white (Cab-o-sil M5) was bought from Cabot Co. Silica sol (40.5 wt% of SiO_2_) was purchased from Qingdao Haiyang Chem. Co., Ltd. Triethylamine, phosphoric acid(85 wt %), sodium aluminate (41 wt% of Al_2_O_3_, 41 wt% of Na_2_O), sodium hydroxide (96 wt%), potassium hydroxide, tetraethylammonium hydroxide (25 wt%), tetrapropylammonium hydroxide(25%) were all obtained from Sinopharm Chemical Reagent Co., Ltd. All the reagents were used as received without further purification.

SAPO-34, SUZ-4, ZSM-5, and MOR zeolites were synthesized following the procedures reported in the literature [23,24,25,26]. USY (Na-FAU, Si/Al = 12) and beta zeolites (Na-BEA, Si/Al = 10) were obtained from commercial manufacturers (XFNANO, NanJing, China). All the as-synthesized zeolites were calcined at 550 °C for 10 h under static air to remove the organic template, followed by two rounds of ion exchange with NH_4_NO_3_ solution (1 M) at 80 °C. Afterwards, the zeolites were washed with deionized water, dried at 110 °C, and then calcined under static air at 550 °C for 10 h to obtain the H-form samples. Further experimental details regarding the characterizations and calculations are provided in the “Appendix A”.

The catalytic performance of zeolites was tested in a stainless steel autoclave lined with Teflon. In each run, DMM (3.8 g), TOM (2.25 g), and catalysts (0.3 g) were loaded into the 25 mL autoclave. The mixture was stirred continuously for 45 min and the reaction temperature was controlled between 30 °C and 150 °C. All products after the reaction consisted of PODE*_n_*, unreacted reactants and byproducts were measured by gas chromatography (GC) and quantitatively analyzed by an internal standard method in which decane was the internal standard. In order to further ensure the accuracy of the results, products were also detected by gas chromatography–mass spectrometry (GC–MS). The conversions of DMM and TOM, denoted as *x*_DMM_ and *x*_TOM_, were calculated by Equations (1) and (2).
*x_DMM_* = (*m*_DMM,feed_ − *m*_DMM,product_)/*m*_DMM,feed_ × 100%(1)
*x*_TOM_ = (*m*_TOM,feed_ − *m*_TOM,product_ − *m*_FA,product_)/*m*_TOM,feed_ × 100%.(2)

The mass selectivity of products was determined by Equation (3)
*M_i_* = m_i,product_/Σ*m*_i_ × 100%.(3)

For example, *M_PODE2−8_* = *m*_PODE2−8,product_/Σ*m*_i_ × 100%, where *m*_PODE2−8_ is the mass of PODE_2−8_ and Σ*m_i_* is the mass of all the trapped liquid products after the reaction.

## 3. Results and Discussion

The X-ray diffraction (XRD) patterns (Appendix A) illustrate that the as-synthesized zeolites were well crystallized with the characteristic peaks of corresponding topology structure [23,24,25,26]. SEM images (Appendix A) further confirm it. The acidities of H-form zeolites were measured by temperature-programmed ammonia desorption (NH_3_-TPD) and infrared spectroscopy of pyridine adsorption (Py-IR) (Figure 2). The Si/Al ratios, surface area, pore volume, detailed weak acid sites (WAS), medium-strong acid sites (MAS), Brønsted acid sites (BAS), and Lewis acid sites (LAS) were calculated (Table 1). Two ammonia desorption peaks were observed in the NH_3_-TPD profiles (Figure 2a), one at 120–250 °C and the other at 250–550 °C, corresponding to WAS and MAS, respectively. The desorption temperature of WAS were generally similar for all the samples; however, there was a notable difference in the desorption temperature of MAS: The higher the desorption temperature, the stronger was the acidic strength. The acidic strength of MAS gradually decreased in the following order: MOR > SUZ-4 > ZSM-5 > SAPO-34 > USY > beta. Furthermore, the peak intensity represents the density of acid sites (Table 1).

The Py-IR of all the zeolites at different temperatures in the region of 1350–1600 cm^−1^ (Appendix A) with the profiles at 250 °C (Figure 2b) were assessed. Two main characteristic bands of pyridine adsorbed on BAS (1540 cm^−1^) and LAS (1450 cm^−1^) were observed [27,28,29]. The amounts of BAS and LAS (Table 1) and their distribution (Figure 2a,b) differ distinctly from the WAS and MAS of the NH_3_-TPD characterization, especially for SUZ-4, MOR, and SAPO-34. The total acid site value of SUZ-4 was 0.840 mmol g^−1^, but only 0.073 mmol g^−1^ for BAS and 0.013 mmol g^−1^ for LAS. The total acid site value of SAPO-34 was 1.07 mmol g^−1^, whereas no BAS or LAS were detected. For MOR, the total acid site value was 1.63 mmol g^−1^ and BAS was only 0.12 mmol g^−1^. This difference is mainly due to variations in zeolite topology structures. SUZ-4 has a channel system that includes 10-membered ring (MR) channels parallel to the unit cell c-axis, which are intersected by two arrays of 8-MR channels running in the plane. SAPO-34 contains large cavities, but these cavities are connected by small 8-MR pore mouths. MOR contains two essentially different channels: Large 12-MR channels and small 8-MR channels that run in parallel and are connected by 8-MR openings. The diameter of 8-MR is much smaller than that of the large pyridine molecule (5.3 Å), and thus, prevents its diffusion into zeolite channels [30]. Therefore, parts of BAS and LAS that are located in 8-MR channels cannot be determined by pyridine adsorption.

A series of preliminary tests with ZSM-5 as a representative catalyst (Appendix A) illustrated that a feed DMM/TOM molar ratio of 2 and a catalyst amount of 5 wt% for 45 min are appropriate for the synthesis of PODE*_n_* [15] and that the zeolite particle size has little effect on catalytic performance. Based on this, the catalytic performance of different zeolites was evaluated as functions of reaction temperature (Figure 3). Beta, ZSM-5, and MOR displayed very similar trends in selectivity to PODE_2−8_ (Figure 3c), reaching a maximum selectivity of 83%–87% in the range of 70–90 °C. Meanwhile, TOM conversion increased quickly before 90 °C and then continued to rise slowly with reaction temperature (Figure 3a). DMM conversion increased initially, reaching a maximum value at 90 °C, and then gradually declined (Figure 3b). The byproduct selectivity of the above three zeolites (Figure 3e−h) were also very similar over the operating temperature range. MeOH and HCHO (FA) decreased, whereas HCOOCH_3_ (MF) and CH_3_OCH_3_ (DME) increased and turned into primary products with increasing reaction temperatures.

In contrast, the catalytic performances of SAPO-34, SUZ-4, and USY exhibited great variability. For SAPO-34 and SUZ-4, a reaction occurred only when the temperature was higher than 100 °C. SAPO-34 exhibited poor catalytic activity, only 47.5% PODE_2−8_ selectivity and 13.8% PODE_2−8_ yield were obtained at 150 °C. SUZ-4 was slightly superior to SAPO-34, with 68.8% PODE_2−8_ selectivity at 110 °C and 48.8% yield at 130 °C. Unlike SAPO-34 and SUZ-4, the catalytic performance of USY gradually increased with a rise of reaction temperature from 30 °C to 150 °C, and ultimately 80.0% PODE_2−8_ selectivity and 72.4% yield were obtained at 150 °C. This result implies that USY can achieve a similar catalytic performance as beta, ZSM-5, and MOR, but only at higher temperatures.

Furthermore, a detailed PODE*_n_* (*n* = 2–11) distribution over USY and ZSM-5 was assessed (Figure 4). As the high Si/Al ratio of HZSM-5 proved effective in the synthesis of PODE*_n_* from DMM and TOM [15], ZSM-5 with a Si/Al ratio of 267 was selected as a research object for comparison. The bar diagrams from bottom to top represent the changing trend of PODE_2_ to PODE_11_ selectivity (Figure 4). The selectivity of PODE*_n_* gradually decreased with the increase of n from 2 to 11 at each reaction temperature. Increasing the Si/Al ratio from 17 to 267 not only increased PODE*_n_* selectivity between 70 °C and 90 °C, but also delayed the rapid selectivity decline at higher temperatures. This indicates that appropriately decreasing the Al content of ZSM-5 could facilitate increasing long-chain PODE*_n_* selectivity and suppressing certain side reactions at high temperatures. Furthermore, USY exhibits poor catalytic performance at low temperatures, but exhibits a similar PODE*_n_* selectivity between 110 °C and 150 °C as ZSM-5 with a high Si/Al ratio.

The synthesis of PODE*_n_* is an acid-catalyzed reaction. Acidic properties, including the density, strength, circumstance, and type of acid sites, directly affect the catalytic performance of zeolites. All the zeolites in this study were relatively similar regarding the desorption temperature of WAS, but exhibited tremendous variations in MAS (Figure 3a). This suggests that catalytic performance strongly depends on MAS. Our previous work has also demonstrated that sufficient Brønsted MAS can effectively promote TOM dissociation and PODE*_n_* chain propagation [15].

Although SAPO-34 contains close to 0.80 mmol g^−1^ MAS, no Brønsted acids were detected that could cause the very low PODE_2−8_ yield. Similarly, SUZ-4 contains only 0.073 mmol g^−1^ Brønsted acids that result in its poor catalytic activity. In comparison, beta, ZSM-5, and MOR exhibit better catalytic performance than SAPO-34 and SUZ-4, which should be attributed to more sufficient BAS. However, it is surprising that USY exhibits completely different catalytic performance that is unlike any other zeolite. The acidic property of USY is very close to beta in regards to acid strength and acid density. The desorption temperature of MAS was approximately 350 °C, and acid density of USY and Beta were 0.24 mmol g^−1^ and 0.23 mmol g^−1^, respectively. Moreover, the Brønsted acid density of USY was more than beta. These results indicate that USY should exhibit similar or better catalytic activity than beta in the same conditions, but obvious differences exist between them. The large differences in catalytic performance of the above-mentioned zeolite catalysts suggest that Brønsted MAS are not the only influential factors in the synthesis of PODE*_n_*. In other words, zeolite acidic properties cannot explain the observed differences in zeolite catalytic performance fully. The following paragraphs will discuss that these differences can be ascribed to differences in channel systems.

Detailed zeolite characteristics (Table 2) were obtained from the International Zeolite Association (IZA) database of zeolite channel systems and related literature [31,32]. Generally, USY, beta, and MOR have 12-ring pores and are usually considered large-pore zeolites; ZSM-5 and SUZ-4 have 10-ring pores and are considered medium-pore zeolites; and SAPO-34 has 8-ring pores and is considered a small-pore zeolite. The maximum included sphere diameters (Table 2) in each zeolite framework type are considered the largest hard sphere or the maximum space available and are usually located inside channel intersections or cage structures and are stationary [32,33].

The pore mouth size of different zeolites is believed to play an important role in the reaction that occurs in the zeolite pore. BAS were not detected in SAPO-34, which we have ascribed to the fact that the pyridine molecule is too large (5.3 Å) to pass through the pore mouth (3.8 Å). This also holds true for the synthesis of PODE*_n_*. Theoretical calculations revealed two conformations of the TOM molecule (Table 3), whose sizes are both larger than 4 Å and thus also larger than the pore mouth size of SAPO-34 (3.8 Å). This indicates that the TOM molecule cannot enter into the SAPO-34 channel and thus subsequent TOM decomposition and PODE*_n_* synthesis cannot occur. That is to say, the small pore mouth size prevents catalytic performance of SAPO-34, while the poor activity of SAPO-34 at high temperatures may be attributed to DMM decomposition or disproportionation, resulting in the formation of PODE*_2_* and other byproducts [33]. This was also illustrated by the DMM disproportionation over H-ZSM-5 at different temperatures (Appendix A). These results suggest that the synthesis of PODE*_n_* occurs in the zeolite pore and that a pore diameter that is larger than a TOM molecule is an essential prerequisite for the reaction.

However, pore mouth size is not the decisive factor resulting in the difference in PODE*_n_* selectivity among USY, ZSM-5, MOR, and beta, whose pore mouth sizes are much larger than the TOM molecule. The maximal included sphere sizes in beta, ZSM-5, MOR, and USY are 6.62 Å, 6.30 Å, 6.64 Å, and 11.18 Å, respectively. The maximum space available is nearly uniform for these zeolites, except USY, which is at least 1.68 times larger than the other three zeolites [33]. This suggests that the maximum included sphere in the zeolite channel may be another key factor. Taking into consideration that the adsorption of reactants onto catalysts is the initial step necessary for the synthesis of PODE*_n_*, models of ZSM-5 and USY were built. The adsorption energy of DMM and TOM and the electrostatic interactions of reacting molecules with the ZSM-5 and USY frameworks were investigated with the density functional theory method. The acid site is located in the intersection of ZSM-5 or the supercage of USY, which provides the maximal space available to accommodate various guest molecules.

The adsorption states of DMM, TOM, and their combination, as well as adsorption energies of ZSM-5 and USY, were modelled (Table 4). ZSM-5 exhibits much higher adsorption energies than USY, regardless of the presence of a single reactant or mixture, suggesting that the interactions between reactants and ZSM-5 frameworks are more intense and that the synthesis of PODE*_n_* should be sensitive to spatial constraints. The reaction of DMM with the Brønsted acidic protons of the zeolite can lead to the formation of methoxymethyl species (CH_3_OCH_2_OZ MM-Z), and TOM can decompose to CH_2_O [15,33]. Celik et al. [33] proposed that the steric constraint of the pore walls in the channel intersections of ZSM-5 leads to high DMM disproportionation rates by promoting the methylene hydrogens of the DMM molecule to interact with the MM-Z. Conversely, the supercage structures of USY allow the hydrogen donor to remain far from the acceptor, so that the driving force for hydrogen transfer is not as strong. The steric constraint of the pore walls may also influence TOM decomposition and PODE*_n_* synthesis in a similar manner. The highest PODE_2−8_ selectivity over ZSM-5 was obtained at 70–90 °C (Figure 3c); however, the main products over USY are MeOH and FA under the same conditions that come from DMM decomposition and TOM decomposition, respectively. This phenomenon may be explained by the fact that the steric constraint of the ZSM-5 channel intersections can preferentially promote the combination of MM-Z and CH_2_O from DMM and TOM to form long-chain ZO(CH_2_O)_n_CH_3_, which further combine with the methyl-end group over Brønsted MAS, resulting in the formation of PODE*_n_*. However, the driving force of USY for the combination of intermediate species is not as strong due to its large size, resulting in poor PODE*_n_* selectivity. This can be further validated by calculating the electrostatic interaction of the DMM and TOM coadsorption complex with the zeolite framework. In comparison to the supercage of USY, the isosurface plots at the intersection of ZSM-5 exhibited a larger green region, which represents the electrostatic stabilization effect (Figure 5). This indicates that the maximum included sphere of ZSM-5 can better stabilize the DMM and TOM coadsorption complex through an appropriate electrostatic stabilization effect, which promotes their further conversion into PODE*_n_* [34,35]. The density functional theory calculation provided further evidence that the catalytic activity and product selectivity of various zeolites in converting DMM and TOM to PODE*_n_* were strongly related to the pore mouth size of the zeolite and the interaction between reactants and zeolite channel systems. The adsorption energy and electrostatic interaction of reacting molecules with SUZ-4, beta, and MOR in Appendix A also proved the above argument.

Another important finding was that increasing reaction temperatures promoted the formation of PODE*_n_* in USY, but rapidly inactivated ZSM-5. This is the result of high temperatures that render intermediate species more active. For USY, the sufficient available space enables a preferential combination of intermediate species from DMM and TOM and restricts side products at high temperatures, resulting in high PODE*_n_* selectivity and very low byproduct selectivity. For ZSM-5 with a Si/Al ratio of 17, the limited space and numerous active acid sites may be more advantageous to TOM decomposition and the subsequent formation of MF by the Tishchenko reaction at high temperatures, while the interaction of DMM with the zeolite and PODE*_n_* synthesis are suppressed. The decomposition of TOM alone over ZSM-5 and USY at 120 °C confirmed this as well. Over HZSM-5, MF was the primary product, whereas over USY, MF almost disappeared in the product, and FA was the dominating product that could take part in the chain propagation forming PODE*_n_* (Appendix A). Increasing the Si/Al ratio of ZSM-5 to 267, i.e., reducing active acid sites (Figure 4), can effectively delay the decline of PODE*_n_* selectivity at high temperatures. This demonstrates that excess active sites and steric constraints of ZSM-5 with a low Si/Al ratio resulted in the enhancement of side reactions at high temperatures.

To sum up, the above facts indicate the obvious synergy between Brønsted MAS and the maximal space available of zeolites for the synthesis of PODE*_n_*. Brønsted MAS make TOM and DMM decompose to monomer CH_2_O and methoxymethoxy groups (MMZ), the steric constraint of the maximum included sphere with an appropriate size in zeolite channels may help to promote the combination of CH_2_O and MMZ to form long-chain ZO(CH_2_O)_n_CH_3_, then ZO(CH_2_O)_n_CH_3_ react with the methyl-end group to form PODE*_n_* over Brønsted MAS [4]. Besides, increasing reaction temperature can increase intermediate species’ activity, so catalytic performance of all studied zeolites is improved gradually before 90 °C. However, too high temperature is not propitious to the formation of PODE*_n_* for low Si/Al ZSM-5, whose excess active acid sites and limited space available may be more adaptable to the conversion of CH_2_O species to MF at higher temperature. On the contrary, high temperature can improve the PODE*_n_* selectivity over USY zeolite due to its excessively large space available.

The stability and recyclability are important indexes for potential industrial application, HZSM-5 and USY were further used to evaluate the reusability for the synthesis of PODE*_n_* at 90 °C. As shown in Figure 6, both HZSM-5 and USY catalyst could exhibit excellent stability and reusability; the catalytic activity only displays a very slight decrease after being reused for 10 cycles upon a simple centrifugation separation. At the same time, the catalytic performance of HZSM-5(Si/Al = 17) catalyst was compared with some recently reported catalysts for the synthesis of PODE*_n_* in Appendix A. When PODE_2__−8_ selectivity and energy consumption are the chief consideration, HZSM-5(Si/Al = 17) showed better catalytic performance than those of many other materials, as illustrated in Appendix A.

## 4. Conclusions

The synthesis of PODE*_n_* from DMM and TOM occurred in the zeolite channel. SAPO-34 contained a supercage structure and numerous strong acid sites; however, the 3.8 Å pore mouth diameter prevented TOM from entering, resulting in very poor activity. Comparing with SAPO-34, SUZ-4 exhibited only a slight advantage in catalytic performance due to very limited Brønsted MAS. ZSM-5, MOR, and beta displayed very similar excellent activity at 70–90 °C, which could be ascribed to sufficient Brønsted MAS and effectual steric constraint of the appropriate sizes of the maximum included spheres. Despite its enough acid sites, USY exhibited lower catalytic activity than other zeolites at low temperatures due to its excessively large included spheres, resulting in weak steric constraint for reactive intermediate species. There was a synergy between Brønsted MAS and the maximal space available of zeolites, which interrelated and jointly influenced the synthesis of PODE*_n_* from DMM and TOM. Brønsted MAS could effectively promote the dissociation of DMM and TOM to MM-Z species and CH_2_O species; the steric constraint of the maximum included sphere with an appropriate size in zeolite channels could promote the combination of MM-Z and CH_2_O to form long-chain ZO(CH_2_O)_n_CH_3_ at lower temperatures; PODE*_n_* were ultimately synthesized by ZO(CH_2_O)_n_CH_3_ reacting with the methyl-end group over Brønsted MAS.

Reaction temperature also affected the synthesis of the PODE*_n_* by changing the activity of intermediate species. High temperature (>90 °C) may lead to more side reactions and suppress the formation of PODE*_n_* for zeolites with limited space, like low Si/Al ZSM-5 zeolite, but can promote PODE*_n_* selectivity for the zeolites with larger space available, like USY zeolite. Moreover, both HZSM-5 and USY catalysts exhibited excellent stability and reusability.

Altogether, a pore mouth diameter larger than the TOM molecule, a proper amount of Brønsted MAS, and an appropriate maximum included sphere size are necessary conditions to obtain high PODE*_n_* selectivity at low temperatures.

## Figures and Tables

**Figure 1 nanomaterials-09-01192-f001:**
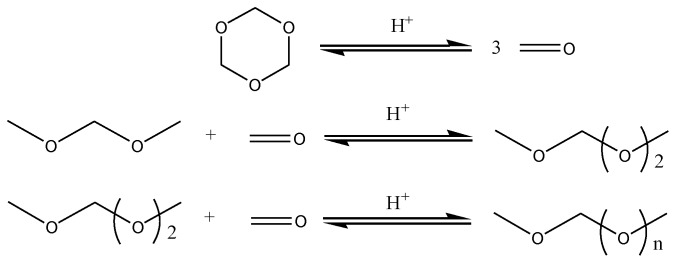
Reaction routes of polyoxymethylene dimethyl ethers (PODE*_n_*) synthesis from dimethoxymethane (DMM) and trioxymethylene (TOM) [4].

**Figure 2 nanomaterials-09-01192-f002:**
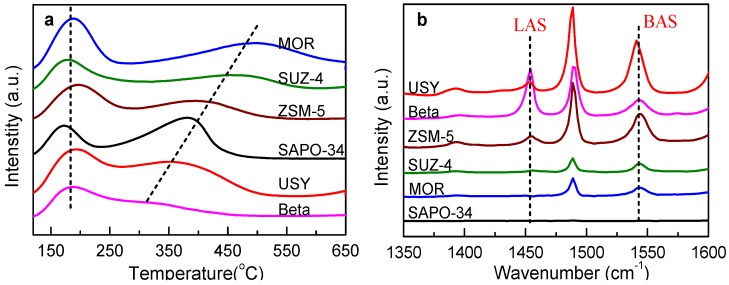
The temperature-programmed ammonia desorption (NH_3_-TPD) profiles (**a**) and pyridine adsorption (Py-IR profiles) (**b**) of various zeolites.

**Figure 3 nanomaterials-09-01192-f003:**
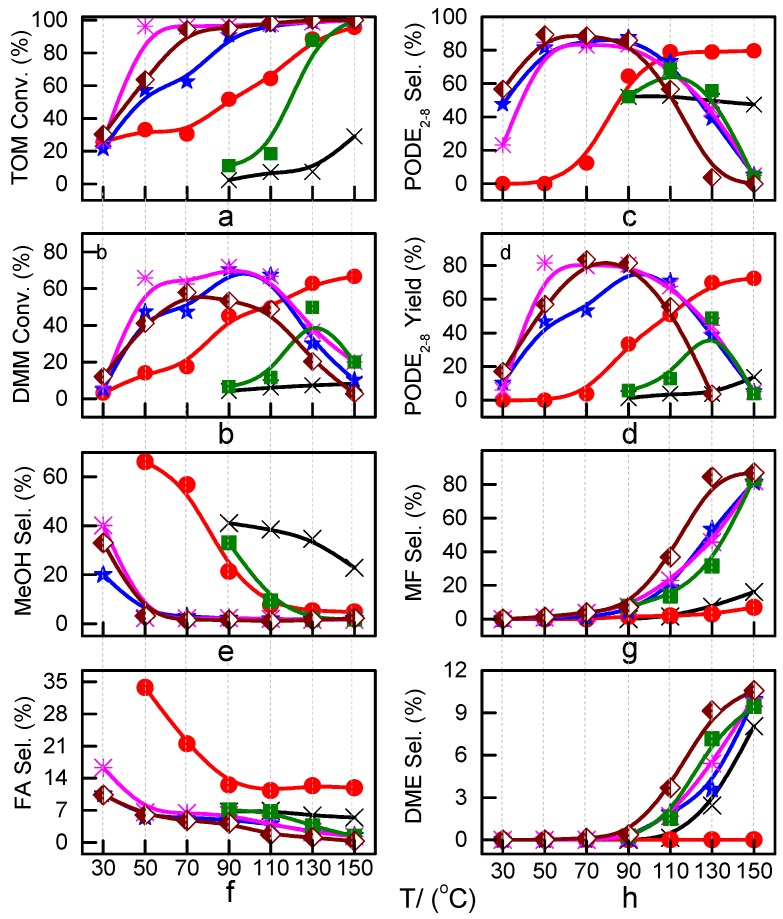
Reactant conversion and product selectivity over different zeolites. 
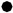
: USY; 
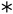
: Beta; 
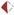
: ZSM-5; ■: SUZ-4; **×**: SAPO-34; 
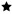
: MOR; (**a**): TOM conversion, (**b**): PODE*_2-8_* selectivity, (**c**): DMM conversion, (**d**): PODE*_2-8_* yield, (**e**): MeOH selectivity; (**f**): FA selectivity; (**g**): MF selectivity; (**h**): DME selectivity.

**Figure 4 nanomaterials-09-01192-f004:**
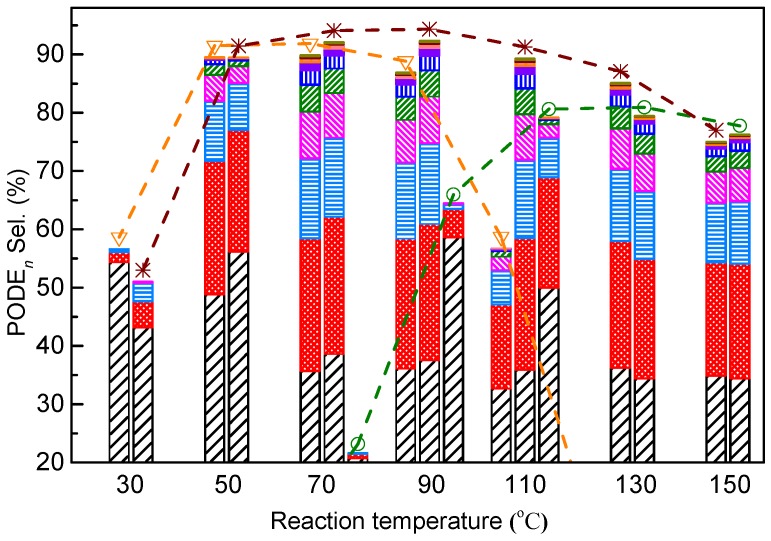
The PODE*_n_* product distribution over ZSM-5 and USY. ▽: ZSM-5 (Si/Al = 17), 
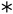
: ZSM-5 (Si/Al = 267), ○: USY (Si/Al = 10), black bar: PODE_2_, red bar: PODE_3_, sky-blue bar: PODE_4_, magenta bar: PODE_5_, olive bar: PODE_6_, blue bar: PODE_7_, violet bar: PODE_8_, orange bar: PODE_9_, wine bar: PODE_10_, dark yellow bar: PODE_11_.

**Figure 5 nanomaterials-09-01192-f005:**
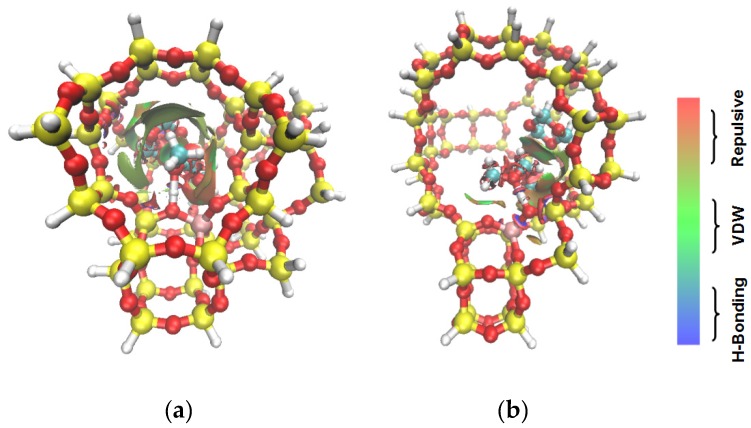
Isosurface plots of the reduced density gradient (s = 0.500 a.u.) for the coadsorption of DMM and TOM in ZSM-5 (**a**) and USY (**b**). The isosurfaces of the reduced density gradient are colored according to the values of the quantity sign (λ2) ρ, as indicated by the RGB scale. VDW: van der Waals interactions.

**Figure 6 nanomaterials-09-01192-f006:**
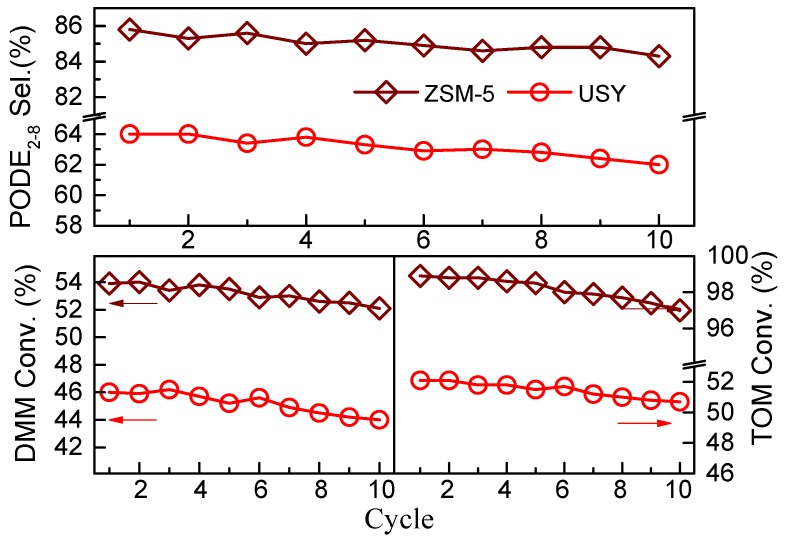
Reusability of the HZSM-5 (Si/Al = 17) and USY (Si/Al = 10) catalyst. The reaction was carried out at 90 °C for 45 min, with a feed DMM/TOM molar ratio of 2 and catalyst amount of 5 wt.%. After each test, the catalyst was reused in the next run following a simple centrifugation separation.

**Table 1 nanomaterials-09-01192-t001:** Physicochemical and acidic properties of various zeolite samples.

Samples	Physicochemical Properties			Acidity by Strength ^d^ (mmol g^−1^)			Acidity by Type ^e^ (mmol g^−1^)	
	Si/Al Ratio ^a^	S_BET_ (m^2^ g^−1^) ^c^	V_pore_ (cm^3^ g^−1^)	Weak	Medium—Strong	Total	Brønsted	Lewis
USY	10	641	0.372	0.40	0.24	0.64	0.50	0.092
Beta	12	500	0.346	0.53	0.23	0.76	0.25	0.40
ZSM-5	17	368	0.231	0.52	0.47	0.99	0.47	0.086
SUZ-4	6	348	0.337	0.40	0.44	0.84	0.073	0.013
SAPO-34	0.085 ^b^	531	0.302	0.27	0.80	1.07	--	--
MOR	15	500	0.259	0.63	1.00	1.63	0.12	0

Notes: ^a^: The Si/Al atomic ratio was determined by inductively coupled plasma atomic emission spectroscopy (ICP-AES); ^b^: This value was obtained by determining the Si/(Al + P) atomic ratio; ^c^: Surface area (S_BET_) and total pore volume (V_pore_) were obtained from nitrogen sorption data; ^d^: density of the acid sites, assorted according to acidic strength, determined by NH_3_-TPD; medium-strong: NH_3_ desorbed at 250–550 °C; weak: NH_3_ desorbed at 120–250 °C; ^e^: Density of the acid sites, assorted according to the acidic type, determined by Py-IR.

**Table 2 nanomaterials-09-01192-t002:** Structural properties of different zeolite framework types.

Samples	Channel System		
	Pore Diameter (nm)	Channel Structure	Maximum Included Sphere Diameter (Å)
USY	0.74 × 0.74	3D, 12-ring	11.18
Beta	0.66 × 0.67	3D, 12-ring	6.62
ZSM-5	0.53 × 0.56	3D, 10-ring	6.30
SUZ-4	0.41 × 0.52	3D, 10-ring	6.27
SAPO-34	0.38 × 0.38	3D, 8-ring	7.31
MOR	0.65 × 0.70	1D, 12-ring	6.64

**Table 3 nanomaterials-09-01192-t003:** The theoretical molecular size of TOM.

TOM Chair Conformation		TOM Boat Conformation	
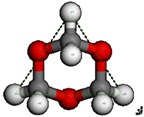		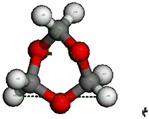	
H4-H6	4.033 Å	H4-H9	4.004 Å
H4-H7	4.033 Å	H8-H5	4.004 Å
H6-H7	4.033 Å	H8-H9	4.038 Å

**Table 4 nanomaterials-09-01192-t004:** Adsorption energy of DMM, TOM, and their combination in ZSM-5 and USY.

Samples	DMM	TOM	DMM + TOM
ZSM-5	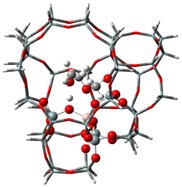	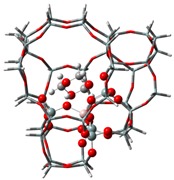	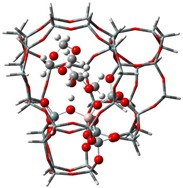
	−90 kJ/mol	−65 kJ/mol	−117 kJ/mol
USY	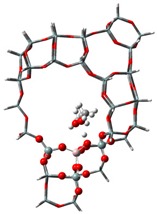	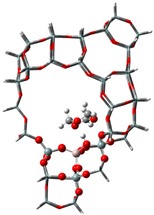	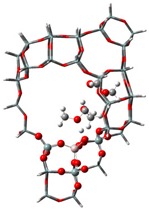
	−60 kJ/mol	−53 kJ/mol	−107 kJ/mol

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
