# Peer review of "The Synergistic Effect of Acidic Properties and Channel Systems of Zeolites on the Synthesis of Polyoxymethylene Dimethyl Ethers from Dimethoxymethane and Trioxymethylene"

_nanomaterials, 2019, doi:10.3390/nano9091192_

Round 1

Reviewer 1 Report

In my opinion, the paper is nice and well written and reports on very interesting and important results. It can be published as it is.

Author Response

Dear reviewer:

     Thanks for your approval to this work. To further improve the manuscript, we made some revisions in the manuscript and these revisions will not influence the content and framework of the paper. Thanks for your recognition again.

Best wishes,

Jianbing Wu

Reviewer 2 Report

The following aspects have to be consider before publication:

- The novelty of the work, as one reads from the Line 23, or 303, appears to be low. All the mentioned factors are reportedly known to contribute to higher selectivity for the considered reaction. So what is realy new? What are exact mechanisms end up determining the selectivity, how they change (if at all) with temperature? Are they they similar for all studied zeolites, or not? Without mentioning/explaining this in the abstract and conclusion parts, the contribution reads as a generic test of different materials for a given reaction under certain conditions with a mediocre scintific input.

- Throughout the manuscript the statements "low activity" and  "high activity" are frequently used without mentioning compared to what (is low or high). I don't believe authors are allowed to define what activity shall be considered as (absolutely) low, and what as high. This must be corrected.

- Similarly, Line 18: "exhibited no catalytic activity". In general, the fact that authors with their instrumentation could not detect any conversion does not mean that there is no activity. Moreover, e.g., Fig. 3a shows about 20% of TOM conversion at 150°C over the SAPO-34. How this can be mentioned as having "no activity"?

- Fig. 4 looks overcrowded and thus is hardly readable. I would recommend to consider a table instead.

- For ammonia-TPD data, the TDP-profile of the reference material used for calculations of the acid site density of studied materials has to be presented (with its own values).

-GC chromatograms based on which conversions and selectivity were calculated should be present in the supporting information.

The following minor remarks are recommended to consider:

Line 57-58: the TOM ring cannot depolymerize to form monomer, since it is not a polymer. Maybe decompose?

Line 91: The equation (2) for TOM conversion appears to be not correct as it contains the mass of formaldehyde. 

Line 98:  Based on what analysis the conclusion of having single phase zeolites is made? Typically, this is done by fitting to the reference XRD patters and reporting the Rietveld refinement numbers. If this is not done, the statements aboth the zeolite purity has to be taken out from the manuscript.

Author Response

Dear reviewer:

Thanks very much for giving us the opportunity to revise our manuscript, and  for the informative comments and instructive revision advices. We have studied comments carefully and made correction which we hope meet with approval. Please see the attachment for details.

Best wishes,

Jianbing Wu

Reviewer 3 Report

The work by Jianbing Wu et al. reports an interesting work on the effect of the acidic properties of some zeolites on the preparation of polyoxymethylene dimethyl ethers. While this work fits the aim and scope of Nanomaterials and is adequate to its wide readership, the research is well accomplished, but discussion suffers from some drawbacks that prevent its recommendation for being accepted at this point.

The manuscript must be revised to address some of the questions raised in the following lines:

The title of the manuscript was not very clear, please change it and make it more appealing. In Figure 2, there is “L acid” and “B acid”, there was another nomenclature for this type of site in the rest of the manuscript. Change to the right nomenclature. In Figure 3 the appearance of the different graphics was not intuitive, probably, it is better to change it. Concerning the catalytic application, authors have conducted it well, but it is necessary to conduct recycling experiments, to show that these new materials are promising catalysts. Authors can conduct it in batch way, when the first cycle ends, add again the amounts of substrates and begin a new catalytic cycle.

In the theoretical calculations there were missing the results for the other materials, add to the manuscript.

After these experiments it is important that authors compare their results with others. This point is very important to show that their materials are better than others or are almost similar to others from literature but presenting advantages.

Based on these facts I am recommending authors to carry out this revision and then the manuscript can be evaluated to be published in Nanomaterials.

Author Response

Dear reviewer:

     Thank you for your comments concerning our manuscript (nanomaterials-562965). These comments are very helpful for revising and improving our paper. We have carefully taken your comments and made correction. Below is our response to your comments.

       Thanks for all the help.

Best wishes,

Jianbing Wu

Round 2

Reviewer 3 Report

Accept in the present form